# Buffer of Thoughts: Thought-Augmented Reasoning with Large Language Models

**Ling Yang**[1][*][†], **Zhaochen Yu**[1][*], **Tianjun Zhang**[2], **Shiyi Cao**[2], **Minkai Xu**[3],
**Wentao Zhang**[1], **Joseph E. Gonzalez**[2], **Bin Cui**[1][†]
[1]Peking University,   [2]UC Berkeley,   [3]Stanford University
Project: https://github.com/YangLing0818/buffer-of-thought-llm
Extension: https://github.com/YangLing0818/SuperCorrect-llm

## Abstract

We introduce Buffer of Thoughts (BoT), a novel and versatile thought-augmented reasoning approach for enhancing accuracy, efficiency and robustness of large language models (LLMs). Specifically, we propose *meta-buffer* to store a series of informative high-level thoughts, namely *thought-template*, distilled from the problem-solving processes across various tasks. Then for each problem, we retrieve a relevant thought-template and adaptively instantiate it with specific reasoning structures to conduct efficient reasoning. To guarantee the scalability and stability, we further propose *buffer-manager* to dynamically update the meta-buffer, thus enhancing the capacity of meta-buffer as more tasks are solved. We conduct extensive experiments on 10 challenging reasoning-intensive tasks, and achieve significant performance improvements over previous SOTA methods: 11% on Game of 24, 20% on Geometric Shapes and 51% on Checkmate-in-One. Further analysis demonstrate the superior generalization ability and model robustness of our BoT, while requiring only 12% of the cost of multi-query prompting methods (e.g., tree/graph of thoughts) on average. Notably, we find that our Llama3-8B + BoT has the potential to surpass Llama3-70B model. Our project is available at https://github.com/YangLing0818/buffer-of-thought-llm

## 1   Introduction

A series of Large Language Models (LLMs) [1–5] like GPT-4 [3], PaLM [2] and LLaMA [6, 7] have showcased the impressive performance in various reasoning tasks. In addition to scaling up the model size to improve the reasoning performance, there are more effective prompting methods that further enhance the functionality and performance of LLMs. We divide these methods into two categories: (i) **single-query reasoning:** these methods [8–10] usually focus on prompt engineering and their reasoning process can be finished within a single query, such as CoT [8] that appends the input query with 'Let's think step by step' to produce rationales for increasing reasoning accuracy, and Few-shot Prompting [11, 12, 9, 13] which provides task-relevant exemplars to assist the answer generation; (ii) **multi-query reasoning:** these methods [14, 15] focus on leveraging multiple LLM queries to elicit different plausible reasoning paths, thus decomposing a complex problem into a series of simpler sub-problems, such as Least-to-Most [16], ToT [14] and GoT [17].

However, both kinds of methods face some limitations: (1) single-query reasoning usually requires prior assumption or relevant exemplars of reasoning process, which makes it impractical to manually design them task by task, thus lacking universality and generalization; (2) Due to the recursive

---

[*]Equal Contribution. Contact: yangling0818@163.com
[†]Corresponding Authors.

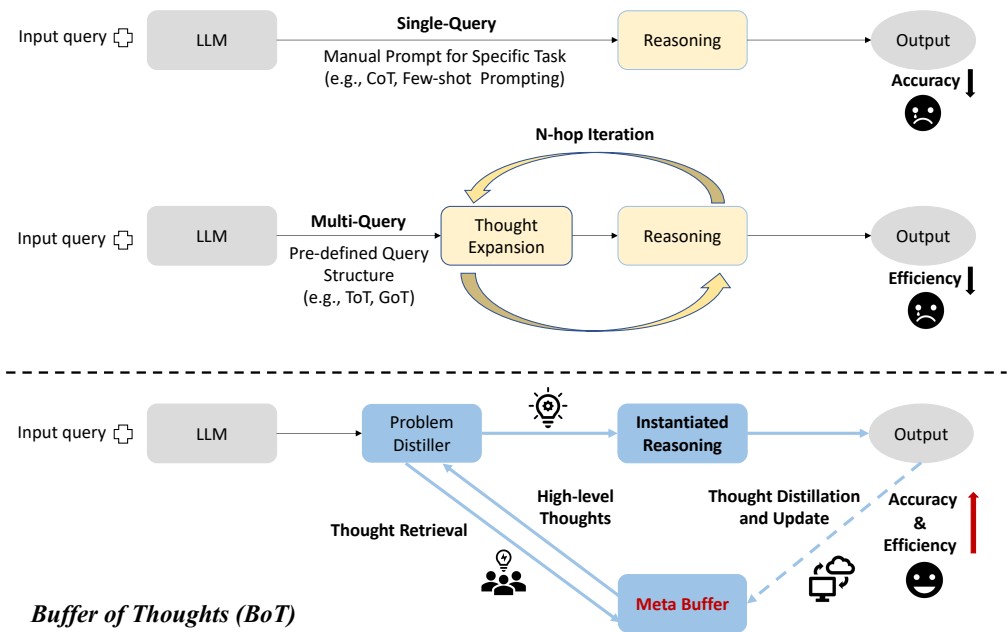

Figure 1: Comparison between single-query [8, 11], multi-query [14, 17], and (c) our BoT methods.

expansion of reasoning paths, multi-query reasoning is usually computationally-intensive when finding a unique intrinsic structure underlying the reasoning process for each specific task; (3) Both single-query and multi-query reasoning processes are limited by their designed exemplars and reasoning structures, and they neglect to derive general and high-level guidelines or thoughts from previously-completed tasks, which are informative for improving efficiency and accuracy when solving similar problems.

To address these limitations, we propose Buffer of Thoughts (BoT), a novel and versatile thought-augmented reasoning framework aimed at enhancing reasoning accuracy, efficiency and robustness of LLMs across various tasks. Specifically, we design *meta-buffer*, a lightweight library housing a series of universal high-level thoughts (*thought-template*), which are distilled from different problem-solving processes and can be shared across tasks. Then, for each problem, we retrieve a relevant thought-template and instantiate it with specific reasoning structure for efficient thought-augmented reasoning. In order to guarantee the scalability and stability of our BoT, we further propose *buffer-manager* to dynamically update the meta-buffer, which effectively enhances the capacity of meta-buffer as more tasks are solved.

Our method has three critical advantages: (i) **Accuracy Improvement:** With the shared thought-templates, we can adaptively instantiate high-level thoughts for addressing different tasks, eliminating the need to build reasoning structures from scratch, thereby improving reasoning accuracy. (ii) **Reasoning Efficiency:** Our thought-augmented reasoning could directly leverage informative historical reasoning structures to conduct reasoning without complex multi-query processes, thus improving reasoning efficiency. (iii) **Model Robustness:** The procedure from thought retrieval to thought instantiation is just like the human thought process, enabling LLMs to address similar problems in a consistent way, thus significantly enhancing the model robustness of our method. Our empirical studies demonstrate that Buffer of Thoughts significantly improves precision, efficiency, and robustness over a diverse array of tasks. Here, we summarize our contributions as follows:

1. We propose a novel thought-augmented reasoning framework Buffer of Thoughts (BoT) for improving the accuracy, efficiency and robustness of LLM-based reasoning.

2. We propose meta-buffer for store informative high-level thoughts distilled from different problems, and adaptively instantiate each thought template to address each specific task.

3. We design buffer-manager to distill thought-templates from various solutions, and is continually improves the capacity of meta-buffer as more tasks are solved.

4. We conduct extensive experiments on 10 challenging reasoning-intensive tasks. Our BoT achieves significant performance improvements over previous SOTA methods: **11% on Game of 24, 20% on Geometric Shapes and 51% on Checkmate-in-One**, while requiring **only 12% of the cost** of multi-query prompting methods on average.

## 2   Related Work and Discussions

**Retrieval-Augmented Language Models**   The retrieval-augmented (Large) Language Model is introduced as a solution to mitigate the phenomenon of hallucination and enhance the output quality of language models [18–22]. When presented with an input question, the retrieval-augmented LLM first queries an external database with billion-level tokens [23] for retrieving a subset of the text corpus to help generating the final answer. Notably, the retrieval-augmented LLM achieves superior question-answering performance using fewer parameters compared to conventional LLMs [19], and it has found application across various downstream tasks [24–26], including multi-modal generation [24, 22, 23, 25] and biomedical applications [26, 27]. In this paper, we construct a novel category of retrieval database, termed *meta-buffer*, which contains a series of high-level thoughts rather than specific instances, aiming to universally address various tasks for LLM-based reasoning.

**Prompt-based Reasoning with Large Language Models**   Prompting techniques have significantly enahnced the arithmetic and commonsense reasoning capabilities of LLMs. Chain-of-Thought (CoT) prompting [8] and its variants [28–30], such as Least-to-Most [16], Decomposed Prompting [31], and Auto-CoT [13]—prompt LLMs to break down complex questions into simpler subtasks and systematically solve them before summarizing a final answer. Numerous studies [32–37] have demonstrated the effectiveness of these prompting methods across a wide range of tasks and benchmarks. Innovations like Tree-of-Thought [14] and Graph-of-Thought [17], have further advanced this field by exploring dynamic, non-linear reasoning pathways to expand heuristic capabilities of LLMs [38, 39]. However, they suffer from increased resource demands and greater time complexity, depend on manual prompt crafting, and are often tailored to specific task types. Recent meta prompting methods [15, 40] utilize a same task-agnostic form of prompting for various tasks and recursively guide a single LLM to adaptively addressing different input queries. Nevertheless, such a long meta prompt may require a considerable context window, and these methods fail to leverage historical informative guidelines or thoughts for potential similar tasks.

**Analogical Reasoning**   Analogical reasoning is a useful technique for natural language reasoning [41–45]. Recent works demonstrate that LLMs can perform analogical reasoning just like humans [46, 47, 12, 48, 49]. For example, Analogical Prompting [12] and Thought Propagation [48] prompt LLMs to self-generate a set of analogous problems, and then utilize the results of analogous problems to produce a solution for input problem. However, the specific solutions for self-explored problems may introduce additional noise and cause error accumulation. Recent Thought-Retriever [49] uses the intermediate thoughts generated when solving past user to address analogous queries, but it only focuses on textual comprehension/generation instead of general reasoning problems. Thus, a more high-level and general analogical approach for LLM complex reasoning is still lacking.

## 3   Buffer of Thoughts

**Overview of Buffer of Thoughts**   In this section, we introduce our Buffer of Thoughts in detail and we also illustrate our core thought-augmented reasoning process in Figure 2. Given a specific task, we utilize our *problem-distiller* (Section 3.1) to extract critical task-specific information along with relevant constraints. Based on the distilled information, we search in *meta-buffer* (Section 3.2) that contains a series of high-level thoughts (*thought-template*) and retrieve a most relevant thought-template for the task. Subsequently, we instantiate the retrieved thought-template with more task-specific reasoning structures and conduct reasoning process. Finally, we employs a *buffer-manager* (Section 3.3) for summarizing the whole problem-solving process and distilling high-level thoughts for imcreasing the capacity of meta-buffer.

### 3.1   Problem Distiller

Most of complex tasks contain implicit constraints, complex object relationships, and intricate variables and parameters within their contexts. Consequently, during the reasoning stage, LLMs need to overcome three main challenges: extracting vital information, recognizing potential constraints, and performing accurate reasoning. These challenges would impose a significant burden on a single LLM. Therefore, we separate the extraction and comprehension stages of task information from the final reasoning stage, through prepending a *problem distiller* to the reasoning process. More

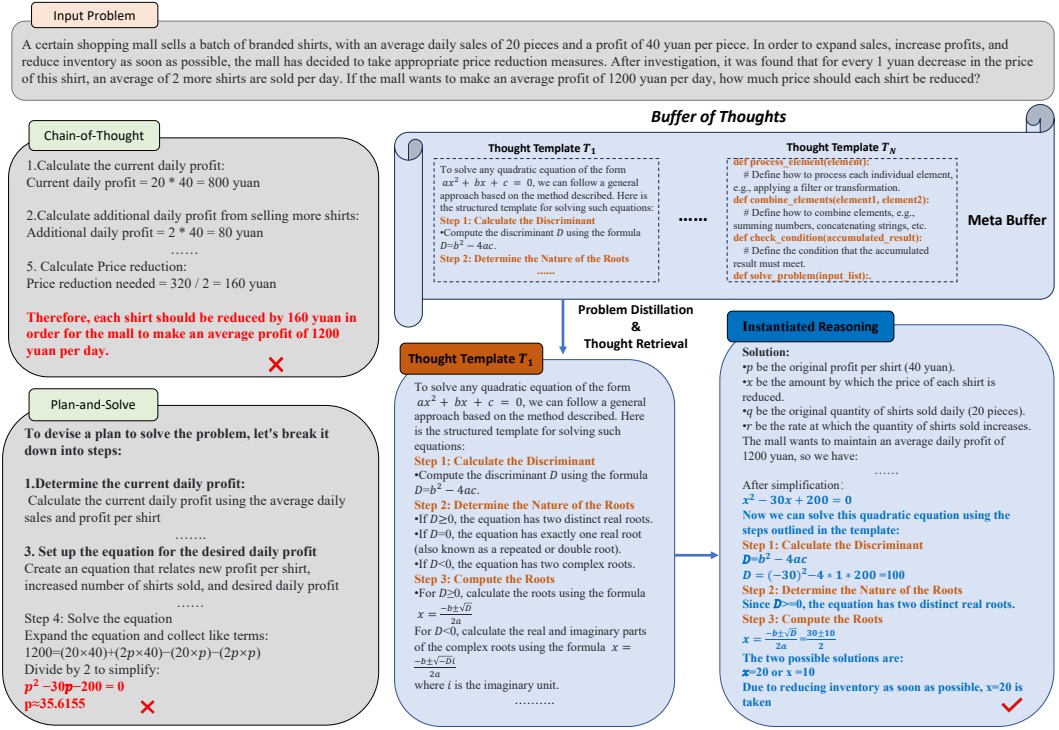

Figure 2: Illustration of different reasoning process. Buffer of Thoughts enables large language models to tackle complex reasoning tasks through our thought-augmented reasoning process. Thought template is marked in orange and instantiated thought is marked in blue.

concretely, we design a meta prompt $\phi$ to first distill and formalize the task information. The distilled task information could be denoted as:

$$x_d = LLM(\phi(x)), \tag{1}$$

where $x$ is the task statement. Due to the page limit, we put the detailed meta prompt for problem-distiller in Appendix B.2.

**Problem Condensation and Translation**    We use the problem distiller to extract key elements from input tasks, focusing on: (1). Essential parameters and variables for problem-solving; (2). The objectives of the input tasks and their corresponding constraints. We then re-organize this distilled information into a clear, comprehensible format for the subsequent reasoning stage. We then translate the specific problems into high-level concepts and structures. This translation procedure decomposes complex real-world problems, like intricate mathematical application scenarios, into simpler, multi-step calculations, making it easier for later retrieval of high-level thought.

## 3.2   Thought-Augmented Reasoning with Meta Buffer

**Motivation**    Human often summarize and induce higher-level guidelines when solving problems and then apply them to relevant problems. Motivated by this, we propose *meta-buffer*, a lightweight library that contains a series of high-level thoughts (*thought-template*) for addressing various types of problems. Unlike traditional methods [11, 46, 12, 36, 9] that require specific instructions or exemplars, our high-level thought-templates can be adaptively instantiated when solving different problems, thereby enhancing LLMs with superior precision and flexibility.

**Thought Template**    As a kind of high-level guideline, our thought-template is stored in meta-buffer , and is obtained from various problem-solving processes by our *buffer-manager*. The details about acquiring thought-templates would be introduced in Section 3.3. Since our BoT aims to provide a general reasoning approach for various tasks, we correspondingly classify the thought-templates into six categories: Text Comprehension, Creative Language Generation, Common Sense Reasoning, Mathematical Reasoning, Code Programming and Application Scheduling. We provide some example thought-templates in Appendix B.1. Such classification of thought-templates can

facilitate the template retrieval for finding most suitable solutions to different problems. Here we denote thought template, template description and its corresponding category as $(T_i, D_{T_i}, C_k)$, where $i$ denotes the index of meta-template, $k \in \mathbb{Z}^+$ and $1 \le k \le 6$, which means $C_k$ is in one of the six categories, and $D_{T_i}$ is the description of thought template.

**Template Retrieval**   For each task, our BoT retrieves a thought-template $T_i$ that is highly similar to the distilled problem $x_d$ by calculating the embedding similarity between the description $D_{T_i}$ and $x_d$. The retrieval process can be formulated as:

$$j = \text{argmax}_i(\text{Sim}(f(x_d), \{f(D_{T_i})\}_{i=1}^N)), \quad \text{where} \quad \text{Sim}(f(x_d), \{f(D_{T_i})\}_{i=0}^n) >= \delta, \quad (2)$$

$N$ is the size of the meta-buffer, $f(\cdot)$ is a normal text embedding model, and $T_j$ denotes the retrieved thought template. We set a threshold $\delta$ (0.5~0.7 is recommended) to determine whether the current task is new. Therefore, if $\text{Sim}(f(x_d), \{f(D_{T_i})\}_{i=0}^n) < \delta$, we identify the task $x$ as a new task.

**Instantiated Reasoning**   For each specific task, we discuss two situations for the instantiated reasoning, depending on whether the current task is new: The first situation is that we successfully retrieve a thought-template $T_j$ for the task. In this case, as presented in Figure 2, our thought-augmented reasoning will be adaptively instantiated to suitable reasoning structures with our designed instantiation prompt (in Appendix B.3). For example, in a Checkmate-in-One problem, we instantiate the template of updating chess board state to solve the problem step by step. Thus we conduct the instantiated reasoning for task $x$ using the distilled information $x_d$ and the retrieved template $T_j$, and produce its solution $S_x$ as:

$$S_x = LLM_{\text{instantiation}}(x_d, T_j), \quad (3)$$

where $LLM_{\text{instantiation}}$ denotes the instantiated reasoner with a LLM.

In the second situation, the task is identified as a new task. To enable proper instantiated reasoning, we prepare three general coarse-grained thought-templates for utilization. Based on the distilled task information $x_d$, our BoT would automatically assign a suitable thought-template to the reasoning process. The detailed pre-defined thought-templates are included in Appendix B.3).

## 3.3   Buffer Manager

We propose *buffer-manager* to summarize the high-level guidelines and thoughts that are gained from each problem-solving process. It can generalize each specific solution to more problems, storing the critical distilled knowledge in the form of thought-templates within the meta buffer. In contrast to methods that **temporarily** generate exemplars or instructions for each problem, our buffer-manager can ensure **permanent** advancements in accuracy, efficiency, and robustness for LLM-based reasoning.

**Template Distillation**   To extract a general though-template, we propose a three-step approach: (1) Core task summarization: identifying and describing basic types and core challenges of problems; (2) Solution steps description: summarize the general steps for solving a problem; (3) General answering template: based on the above analysis, propose a solution template or approach that can be widely applied to similar problems. Additionally, to boost the generalization ability and stability of template distillation, we carefully design two types of in-context examples of how to generate thought-template—*in-task* and *cross-task* examples. *Cross-task* means we choose the template distilled from one task to tackle the problem of other tasks, such as addressing a mathematical problem with a code-related thought-template. The new template distilled from input task $x$ can be denoted as:

$$T_{new} = LLM_{\text{distill}}(x_d, S_x), \quad (4)$$

where $LLM_{\text{distill}}$ is the LLM-based template distiller initialized with the following prompt:

> ***Prompt for Template Distillation:***
> **User: [Problem Description] + [Solution Steps or Code]**
> *To extract and summarize the high-level paradigms and general approaches for solving such problems, please follow these steps in your response:*
> **1. Core task summarization:**
> *Identify and describe the basic type and core challenges of the problem, such as classifying it as a mathematical problem (e.g., solving a quadratic equation), a data structure problem (e.g., array sorting), an algorithm problem (e.g., search algorithms), etc. And analyze the most efficient way to solve the problem.*
> **2. Solution Steps Description:**
> *Outline the general solution steps, including how to define the problem, determine variables, list key equations or constraints, choose appropriate solving strategies and methods, and how to verify the correctness of the results.*
> **3. General Answer Template:**
> *Based on the above analysis, propose a template or approach that can be widely applied to this type of problem, including possible variables, functions, class definitions, etc. If it is a programming problem, provide a set of base classes and interfaces that can be used to construct solutions to specific problems.*
> *Please ensure that your response is highly concise and structured, so that specific solutions can be transformed into generalizable methods.*
> ***[Optional] Here are some exemplars of the thought-template:*** *(Choose cross-task or in-task exemplars based on the analysis of the* **Core task summarization**.*)*

**Dynamic Update of Meta-Buffer**    After template distillation, we need to consider whether the distilled template should be updated into the meta-buffer. If we initialize an empty meta-buffer or encounter a problem without a proper thought-template, the distilled thought-templates will be directly stored in the meta-buffer. If we solve problem with a retrieved thought-template, new insights may arise during the instantiation of a certain thought-template. Therefore, to avoid the redundancy of the meta-buffer while maintaining newly-generated informative thoughts, we will calculate the similarity between the embedding vectors of $D_{T_{new}}$ and $\{D_{T_i}\}_{i=0}^n$ and update the meta-buffer with the following rule:

$$\text{Max}(\text{Sim}(f(D_{T_{new}}), \{f(D_{T_i})\}_{i=0}^n)) < \delta. \tag{5}$$

Otherwise, it means the meta-buffer has already possessed the necessary knowledge to solve this task and does not need to perform the update. Our dynamic update strategy effectively reduces the computational burden of template retrieval while ensuring the lightweight property of our meta-buffer. We further conduct ablation study to analyze it in Section 4 and Appendix A.

## 4   Experiments

**Datasets and Tasks**    To evaluate the efficacy of our proposed Buffer of Thoughts and compare with previous methods, we consider a diverse set of tasks and datasets that require varying degrees of mathematical and algorithmic reasoning, domain-specific knowledge, and literary creativity: (a). The **Game of 24** from ToT [14], where the objective is to form an arithmetic expression that equals 24 using each of four given numbers exactly once; (b). Three BIG-Bench Hard (BBH) [35] tasks: **Geometric Shapes**, **Multi-Step Arithmetic Two**, and **Word Sorting**; (c). Three reasoning tasks directly obtained from the BIG-Bench suite [50]: **Checkmate-in-One**, **Penguins**—where the task is to answer questions about penguins' attributes based on a given table and additional natural language information, and **DateUnderstanding**—a task that involves inferring dates from natural language descriptions, performing arithmetic operations on dates, and utilizing global knowledge such as the number of days in February; (d). **Python Programming Puzzles** (P3) [51, 52], a collection of challenging programming puzzles written in Python with varying difficulty levels; (e). **Multilingual Grade School Math** (MGSM) [33], a multilingual version of the GSM8K dataset [53] featuring translations of a subset of examples into ten typologically diverse languages, including Bengali, Japanese, and Swahili; (f). **Shakespearean Sonnet Writing** from meta-prompting [15], a novel task where the goal is to write a sonnet following the strict rhyme scheme "ABAB CDCD EFEF GG" and incorporating three provided words verbatim.

Table 1: Comparing BoT with previous methods across various tasks. We denote the best score in blue, and the second-best score in green. Our BoT significantly outperforms other methods on all tasks, especially on general reasoning problems.

| Task | Standard | Single-Query | | | Multi-Query | | | BoT (Ours) |
|------|----------|--------------|--|--|-------------|--|--|------------|
| | GPT4 [3] | GPT4+CoT [8] | Expert [9] | PAL [10] | ToT [14] | GoT [17] | Meta Prompting [15] | |
| Game of 24 | 3.0 | 11.0 | 3.0 | 64.0 | 74.0 | 73.2 | 67.0 | 82.4 |
| MGSM (avg) | 84.4 | 85.5 | 85.0 | 72.0 | 86.4 | 87.0 | 84.8 | 89.2 |
| Multi-Step Arithmetic | 84.0 | 83.2 | 83.2 | 87.4 | 88.2 | 89.2 | 90.0 | 99.8 |
| WordSorting | 80.4 | 83.6 | 85.2 | 93.2 | 96.4 | 98.4 | 99.6 | 100.0 |
| Python Puzzles | 31.1 | 36.3 | 33.8 | 47.3 | 43.5 | 41.9 | 45.8 | 52.4 |
| Geometric Shapes | 52.6 | 69.2 | 55.2 | 51.2 | 56.8 | 54.2 | 78.2 | 93.6 |
| Checkmate-in-One | 36.4 | 32.8 | 39. 6 | 10.8 | 49.2 | 51.4 | 57.2 | 86.4 |
| Date Understanding | 68.4 | 69.6 | 68.4 | 76.2 | 78.6 | 77.4 | 79.2 | 88.2 |
| Penguins | 71.1 | 73.6 | 75.8 | 93.3 | 84.2 | 85.4 | 88.6 | 94.7 |
| Sonnet Writing | 62.0 | 71.2 | 74.0 | 36.2 | 68.4 | 62.8 | 79.6 | 80.0 |

**Implementation and Baselines** For the fair comparisons with previous methods, we use GPT-4 as the base model of our BoT, including the main experiment and the ablation study. We also use Llama3-8B and Llama3-70B in our analysis part on NVIDIA A100-PCIE-40GB GPU. We compare our Buffer of Thoughts with the following prompting methods: **1. Standard Prompting**: This is our most basic baseline, where an LLM is asked to generate a response directly from the input query, without any specific guiding input-output examples or additional instructions beyond the task description included in the query.

**2. Single-query Method**: This includes Zero-shot CoT [8] and PAL [10], which use the LLM to analyze natural language problems and generate intermediate reasoning steps. We also include Expert Prompting [9], which creates an expert identity tailored to the specific context of the input query, and then integrates this expert profile into the input to generate a well-informed response.

**3. Multi-query Method**: This includes ToT [14] and GoT [17], which enable LLMs to make deliberate decisions by considering multiple reasoning paths and self-evaluating choices to determine the next course of action. These methods also allow for looking ahead or backtracking when necessary to make global decisions. Additionally, we include Meta Prompting [15], which employs an effective scaffolding technique designed to enhance the functionality of LLMs.

### 4.1 BoT Achieves Better Accuracy, Efficiency and Robustness

**Reasoning Accuracy** As shown in Table 1, our BoT consistently outperforms all previous prompting methods across multiple kinds of challenging benchmarks, particularly demonstrated in complicated reasoning tasks such as Game of 24 and Checkmate-in-One. Taking GPT-4 as a baseline, our method achieves an astonishing 79.4% accuracy improvement in Game of 24, and compared to ToT, which has a good performance on this task, we also achieve an 8.4% accuracy improvement. What's more, compared to recent Meta-prompting method [15], we see **significant accuracy improvements: 23% on Game of 24, 20% on Geometric Shapes and 51% on Checkmate-in-One**. Existing methods need complex, iterative, and heuristic search strategies to address these problems on a case-by-case basis. Conversely, our BoT leverages the historical insights and informative guidelines from thought-templates, and further adaptively instantiate a more optimal reasoning structure for addressing these complex problems.

**Reasoning Efficiency** In addition to significant improvements in accuracy, as a multi-query method, our BoT can achieve comparable reasoning time to single-query method across various tasks, while being considerably less than conventional multi-query method like ToT [14] as shown in Figure 3. For example, in Game of 24, both single-query and multi-query methods necessitate iterative and heuristic searches to identify feasible solutions. This process is particularly time-consuming and inefficient, especially for the multi-query method, which involves conducting multi-query search and backtrace phases. In contrast, our BoT directly retrieves a thought-template in code format, thus a program is instantiated to traverse combinations of numbers and symbols, thereby eliminating the need to build the reasoning structure from scratch. This allows for solving the problem with just one query after invoking the problem-distiller, significantly reducing the time required for complex reasoning. Notably, our **BoT requires only 12% of the cost of multi-query methods** (e.g., tree of thoughts and meta-prompting) on average.

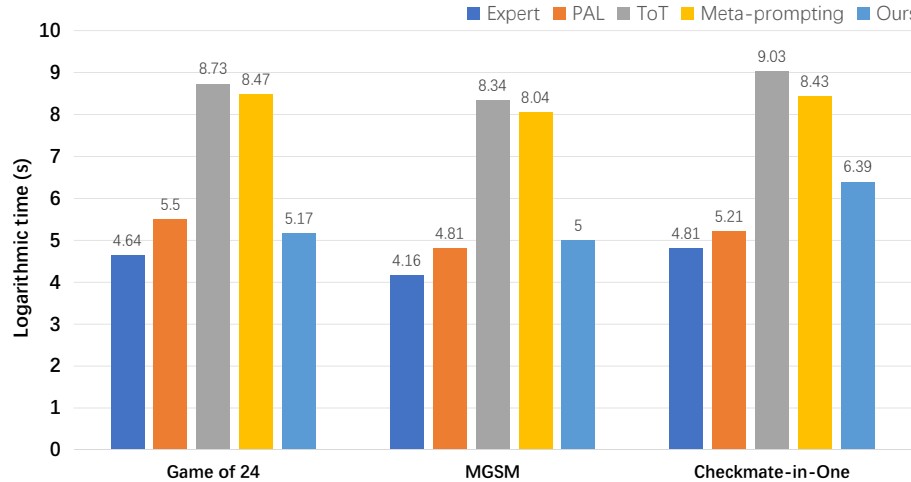

Figure 3: Comparison of **logarithmic inference time** between our Buffer of Thoughts and GPT4 [3], GPT4+CoT [8], Expert-prompting [9], PAL [10], ToT [14] across different benchmarks.

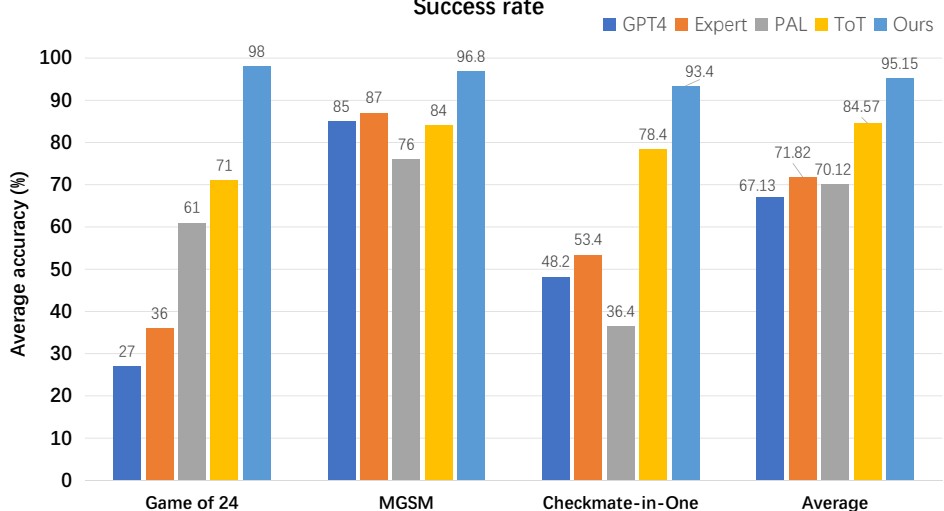

Figure 4: Comparison of reasoning robustness between our Buffer of Thoughts and GPT4 [3], GPT4+CoT [8], Expert-prompting [9], PAL [10], ToT [14] across different benchmarks.

**Reasoning Robustness**    To better evaluate our BoT, we devise a new evaluation metric: *success rate*, which is used to assess the reasoning robustness. We randomly sample 1000 examples from various benchmarks as a test subset and evaluate different methods on this subset. As shown in Figure 4, we repeat this evaluation process 10 times and take the average accuracy as the success rate of different methods on each benchmark. Compared with other methods, our BoT consistently maintains a higher success rate across various tasks, surpassing the second-best by 10% in average success rate. We attribute our outstanding robustness to the great generalization ability of our distilled thought-templates during reasoning across different tasks. By offering high-level thought from the suitable thought-templates, the stability of our method across different tasks is greatly enhanced.

## 5   Model Analysis

**Distribution Analysis of Thought-Templates**    As depicted in the left figure of Figure 5, we choose six different benchmarks, each sampled with 100 distinct tasks. We update the meta-buffer from scratch, and after completing all sampled tasks, we display the number of derived thought-templates. We can observe that our BoT generates a greater number of thought-templates in the MGSM tasks that contain more diverse scenarios. In tasks with relatively simple requirements, such as Checkmate-in-One and Penguins, BoT produces more fixed thought-templates tailored for those specific issues. The distribution of templates indicates that our BoT can effectively discover appropriate thought templates for different benchmarks.

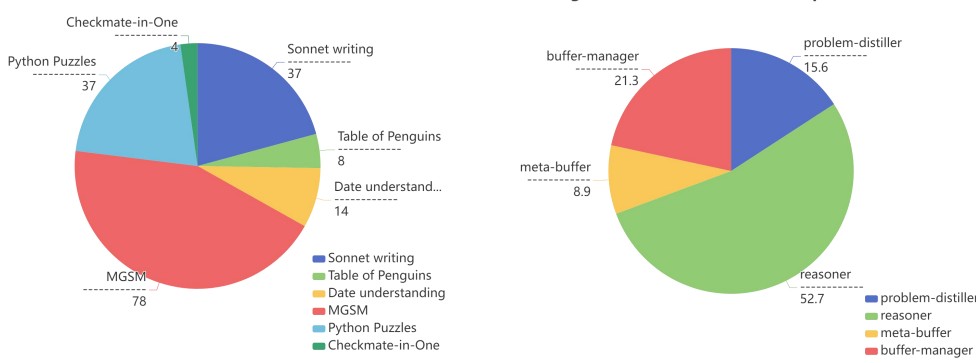

Figure 5: Distribution Analysis of Thought-Templates and Time. *Left*: Distribution Analysis of Thought-Templates. *Right*: Time Distribution of BoT.

**Distribution Analysis of Time Cost**    As illustrated in Figure 5, we measured the average time cost for each component of BoT's reasoning framework across different tasks. The time required for distilling task information and template retrieval is relatively short, whereas instantiated reasoning takes longer. Overall, considering the complexity of different components, our BoT achieves a relatively balanced distribution of time cost, demonstrating the efficiency of our BoT framework.

**Better Trade-off between Model Size and Performance**    As depicted in Figure 6, on Game of 24, word list sorting and Checkmate-in-One, Llama3-8B and Llama-70B models [6] may result in poor outcomes. However, equipped with our BoT, both models demonstrate a substantial accuracy improvement. Notably, **BoT+Llama3-8B has the potential to surpass single Llama3-70B model**. Our BoT enables smaller models to exhibit the capabilities that approximate or even surpass larger models, significantly bridging the gap between their reasoning abilities. Furthermore, it greatly diminishes the inference cost required by large language models when tackling complex problems.

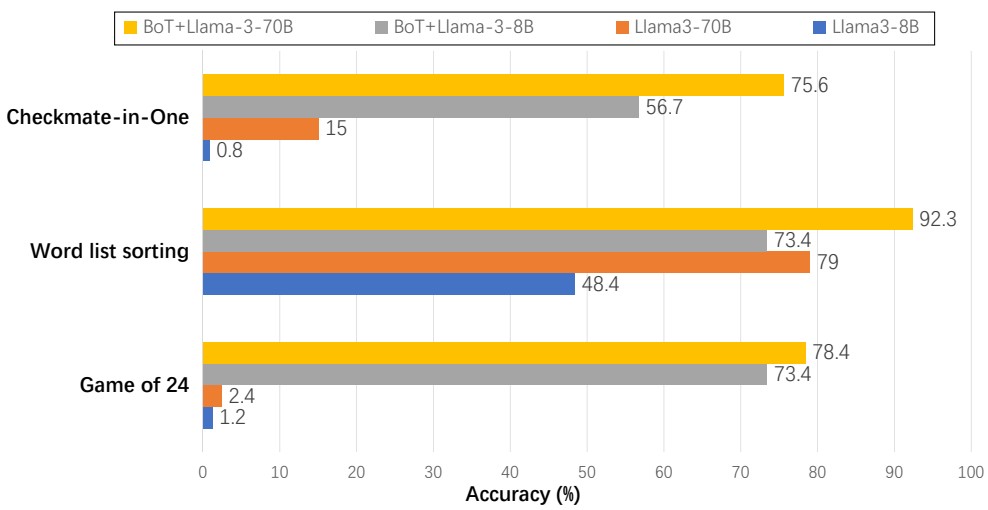

Figure 6: We evaluate the trade-off between model size and performance with Llama3-8B and Llama3-70B models on three challenging benchmarks.

**Quanlity of Automatically-Induced Template**    The success of the proposed approach critically depends on the quality of the automatically induced template. While this previous experiments has shown promising empirical performance on downstream tasks, it remains unclear how good the templates themselves are. Thus we make a comparison between the automatically generated task templates with manually prepared templates for more complex reasoning tasks on MATH dataset [54], with randomly sampled 500 problems. From the results, we can find that our automatic template boosts the reasoning ability of LLMs, demonstrating its generalization ability.

**Impact of Buffer-Manager**    We further conduct ablation study on our buffer-manager, where we divide the entire process into four rounds. In each round, we randomly sample 50 questions from

Table 2: Compare our automatic thought templates with manual ones with GPT-3.5 on MATH.

| Accuracy | Manual Template | Automatic Template | Automatic Template (after accumulation) |
|---|---|---|---|
| MATH-500 | 52.8% | 73.4% | 78.4% |

each benchmark and conduct reasoning. In the subsequent round, we continue to randomly sample another 50 questions from each benchmark. As depicted in Figure 7, with the increase of the number of rounds, the model with the buffer-manager continually expands the meta-buffer while also utilizing the thought-templates obtained from previously solved problems to help addressing subsequent similar problems. Therefore, we can observe that the accuracy of BoT steadily improves with each round. In contrast, the model without the buffer-manager fails to exhibit an upward trend.

Additionally, we also demonstrate the superiority of our buffer-manager on the reasoning efficiency as depicted in Figure 10. when the number of rounds increases, the model with the buffer-manager will experience a continual improvement in reasoning efficiency. This is because, with the continual expansion of the meta-buffer, the likelihood of retrieving suitable thought-templates also increases. Consequently, models can avoid constructing reasoning structures from scratch, thereby enhancing the inference efficiency accordingly. More ablation study can be found in Appendix A.

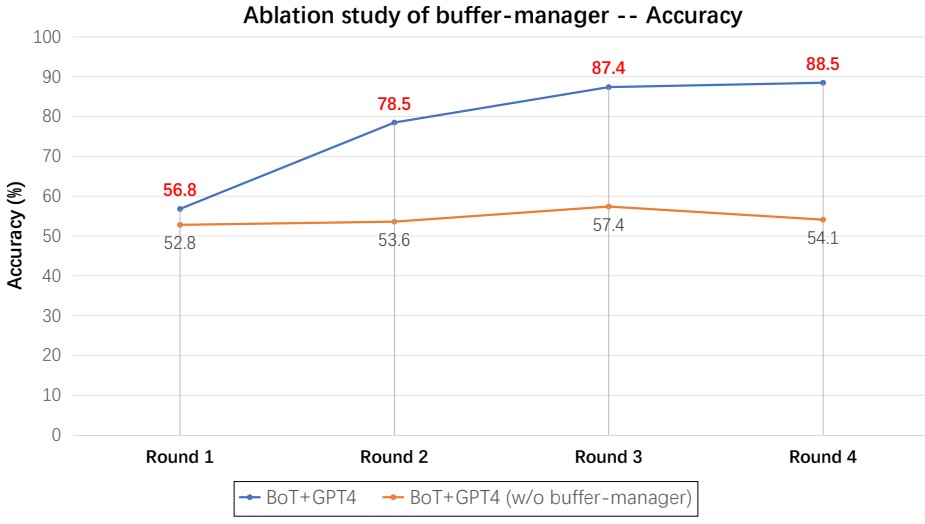

Figure 7: We conduct ablation study on buffer-manager regarding reasoning accuracy across four tasks, employing Llama3-70B and GPT-4 as the base models.

## 6 Discussion

In this work, we introduce Buffer of Thoughts, a novel beffered reasoning framework that employs LLMs to utilize pre-accumulated experiences and methodologies from prior tasks for progressively raising the LLM's reasoning capacity. our BoT brings out a set of future directions: (1). integrating external resources with BoT to build a open-domain system like agent models [55, 56]. (2). making the distillation of thought-templates optimizable, which may significantly enhance their template qualities for more complex tasks. (3). incorporating BoT into LLM training for eliciting more fine-grained and accurate reasoning process, like SuperCorrect [57].

## Acknowledgements

This work is supported by National Natural Science Foundation of China (U23B2048, U22B2037), Beijing Municipal Science and Technology Project (Z231100010323002), research grant No. SH-2024JK29, the Fund of Kunpeng and Ascend Center of Excellence (Peking University), and High-performance Computing Platform of Peking University.

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

# A    More Ablation Studies

## A.1    Impact of Problem-Distiller

As illustrated in Figure 8, when the problem-distiller is disabled, both Llama3-70B and GPT-4 experience a certain degree of accuracy decline. More complex problems, such as Game of 24 and Checkmate-in-One, show a more significant accuracy reduction, whereas relatively simpler problems like word list sorting and MGSM exhibit smaller decreases. This is because LLMs can more easily extract key information in simpler tasks, making the impact of the problem-distiller less noticeable. In contrast, extracting key information and potential constraints in complex problems is more challenging, making the role of our problem-distiller more prominent, thereby explaining the differences depicted in the figure.

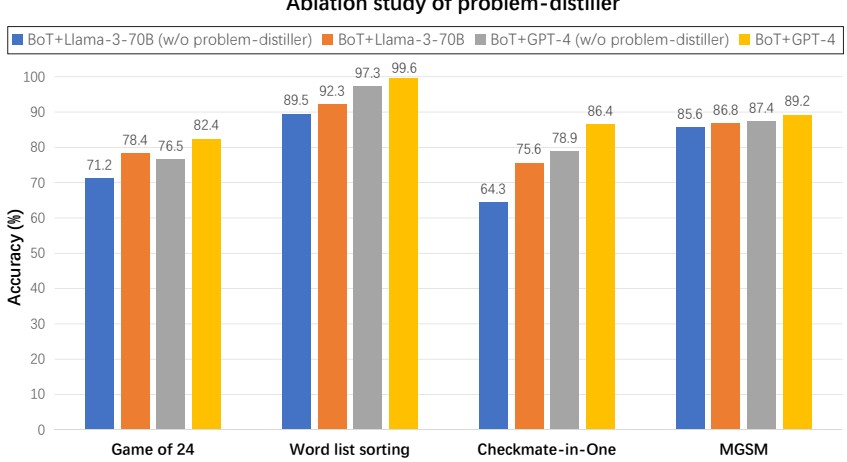

Figure 8: We conduct ablation study on problem-distiller across four benchmarks, employing Llama3-70B and GPT-4 as the base models.

## A.2    Impact of Meta-Buffer

As illustrated in Figure 9, when the meta-buffer is disabled, both Llama3-70B and GPT-4 models exhibit a noticeable decline in performance, particularly in benchmarks requiring complex reasoning, such as Game of 24 and Checkmate-in-One. This further underscores the superiority of our meta-buffer in addressing complex problems. We would extend such mechanism to more scenarios [57–61].

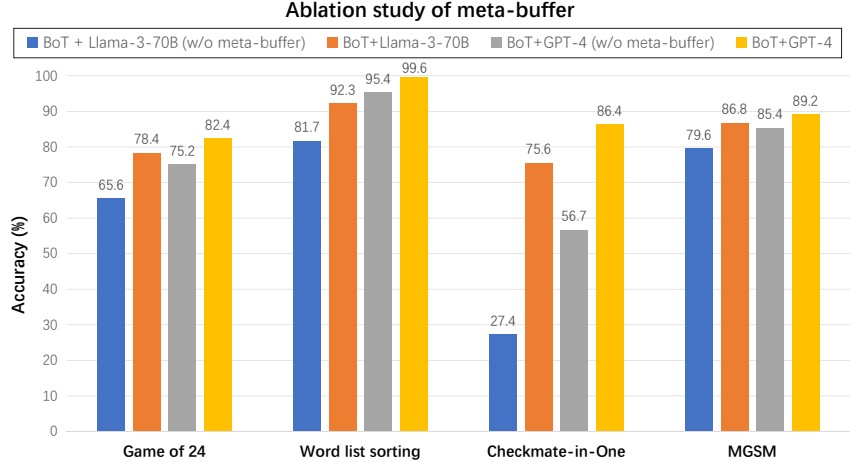

Figure 9: We conduct ablation study on meta-buffer across four benchmarks, employing Llama3-70B and GPT-4 as the base models.

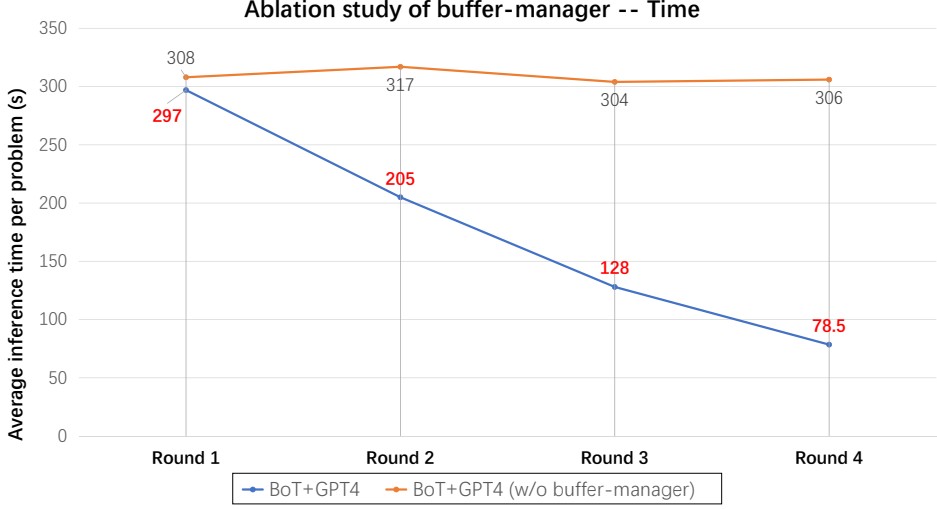

Figure 10: We conduct ablation study on buffer-manager regarding reasoning efficiency across four tasks, employing Llama3-70B and GPT-4 as the base models.

# B  Additional Method Details

## B.1  Six Kinds of Detailed Thought-Templates

**1. Text Comprehension**

**Task Description:**
The task involves analyzing a table with various attributes of penguins, such as name, age, height, and weight, and answering questions about these attributes. The table may be updated with new entries, and additional context or comparisons may be provided in natural language.

- - - - - - - - - - - - - - - - - - - - - - - - - - - - - - - - - - - - - - - - - - -

**Solution Description:**
To accurately answer questions about the penguins' attributes, one must be able to interpret the data presented in tabular form, understand any additional information provided in natural language, and apply logical reasoning to identify the correct attribute based on the question asked.

**Thought Template:**
Step 1: Parse the initial table, extracting the header information and each penguin's attributes into a structured format (e.g., a list of dictionaries).
Step 2: Read and integrate any additional natural language information that updates or adds to the table, ensuring the data remains consistent.
Step 3: Identify the attribute in question (e.g., oldest penguin, heaviest penguin) and the corresponding column in the table.
Step 4: Apply logical reasoning to compare the relevant attribute across all entries to find the correct answer (e.g., the highest age for the oldest penguin).
Step 5: Select the answer from the provided options that matches the result of the logical comparison.

## 2. Creative Language Generation

**Task Description:**
The task is to generate a sonnet that adheres to the traditional English sonnet rhyme scheme of "ABAB CDCD EFEF GG" and includes three specific words verbatim in the text.

- - - - - - - - - - - - - - - - - - - - - - - - - - - - - - - - - - - - - - - - - - - - - - - - -

**Solution Description:**
Writing a sonnet involves crafting 14 lines of poetry that follow a specific rhyme pattern. The lines are typically in iambic pentameter, though flexibility in rhythm can be allowed for creative reasons. The given rhyme scheme dictates the end sounds of each line, ensuring a structured poetic form. Incorporating the three provided words verbatim requires strategic placement within the lines to maintain the poem's coherence and thematic unity.

**Thought Template:**
Step 1: Identify the three words that must be included in the sonnet.
Step 2: Understand the rhyme scheme "ABAB CDCD EFEF GG" and prepare a list of rhyming words that could be used.
Step 3: Develop a theme or story for the sonnet that can naturally incorporate the three provided words.
Step 4: Begin drafting the sonnet by writing the first quatrain (four lines) following the "ABAB" rhyme scheme, ensuring one or more of the provided words are included.
Step 5: Continue with the second quatrain "CDCD," the third quatrain "EFEF," and finally the closing couplet "GG," each time incorporating the provided words as needed.
Step 6: Review the sonnet for coherence, flow, and adherence to the rhyme scheme, making adjustments as necessary.

## 3. Common Sense Reasoning

**Task Description:**
Given a specific date and an event, such as a holiday or historical event, determine the following date.

- - - - - - - - - - - - - - - - - - - - - - - - - - - - - - - - - - - - - - - - - - - - - - - - -

**Solution Description:**
To determine the next date, we need to consider the structure of the calendar, the number of days in each month, and whether it's a leap year. Typically, the number of days in a month is fixed, except February may vary due to leap years. The next day in a year is usually the date increased by one day unless it's the end of the month, then the next day will be the first day of the following month. For the end of the year, the next day will be January 1st of the following year.

**Thought Template:**
Step 1: Identify the given date's month and day number.
Step 2: Check if it's the end of the month; if so, confirm the start date of the next month.
Step 3: If it's not the end of the month, simply add one to the day number.
Step 4: Pay special attention to the end of the year, ensuring the year increments.

## 4. Code Programming

**Task Description:**
When given a list of numbers, try to utilize 4 basic mathematical operations (+-*/) to get a target number.

**Thought Template:**

Listing 1: Python template

```python
from itertools import permutations, product

def perform_operation(a, b, operation):
    # Define the operation logic (e.g., addition, subtraction,
        etc.).
    pass

def evaluate_sequence(sequence, operations):
    # Apply operations to the sequence and check if the result
        meets the criteria.
    pass

def generate_combinations(elements, operations):
    # Generate all possible combinations of elements and
        operations.
    pass

def format_solution(sequence, operations):
    # Format the sequence and operations into a human-readable
        string.
    pass

def find_solution(input_elements, target_result):
    # Data Input Handling
    # Validate and preprocess input data if necessary.

    # Core Algorithm Logic
    for sequence in permutations(input_elements):
        for operation_combination in generate_combinations(
            sequence, operations):
            try:
                if evaluate_sequence(sequence,
                    operation_combination) == target_result:
                    # Data Output Formatting
                    return format_solution(sequence,
                        operation_combination)
            except Exception as e:
                # Error Handling
                # Handle specific exceptions that may occur
                    during evaluation.
                continue

    # If no solution is found after all iterations, return a
        default message.
    # return No solution found message
    return

# Example usage:
input_elements = [1, 7, 10, 3]
target_result = 24
print(find_solution(input_elements, target_result))
```

**5. Application Scheduling**

**Task Description:**
Given some Chess moves in SAN, update the chess board state.

Listing 2: Python template

```python
import chess
def find_checkmate_move(moves_san):
    # Initialize a new chess board
    board = chess.Board()

    # Apply the moves to the board
    for move_san in moves_san:
        # Remove move numbers and periods (e.g., "1." or "2.")
        if len(move_san.split('. ')) > 1:
            move_san = move_san.split('. ')[1]
        # Skip empty strings resulting from the removal
        if move_san:
            # Apply each move in SAN format to the board
            move = board.parse_san(move_san)
            board.push(move)

    # Generate all possible legal moves from the current
        position
    for move in board.legal_moves:
        # Make the move on a copy of the board to test the
            result
        board_copy = board.copy()
        board_copy.push(move)

        # Check if the move results in a checkmate
        if board_copy.is_checkmate():
            # Return the move that results in checkmate in SAN
                format
            return board.san(move)
    # return No solution found message
    return

#Example usage:
input = '......'
# Check input format and transform the input into legal format
# Remove move numbers and periods (e.g., "1." or "2.")
checkmate_move = find_checkmate_move(moves_san)
print(checkmate_move)
```

## 6. Mathematical Reasoning

**Task Description:**
Solve an quadratic equation of the form $ax^2 + bx + c = 0$ considering any situations.

- - - - - - - - - - - - - - - - - - - - - - - - - - - - - - - - - - - - - - - - - - - - - - - - - - - -

**Solution Description:**
To solve any quadratic equation of the form $ax^2 + bx + c = 0$, we can follow a general approach based on the method described. Here is the structured template for solving such equations:

**Thought Template:**
Step 1: Calculate the Discriminant
- Compute the discriminant $D$ using the formula $D = b^2 - 4ac$.
Step 2: Determine the Nature of the Roots
- If $D > 0$, the equation has two distinct real roots.
- If $D = 0$, the equation has exactly one real root (also known as a repeated or double root).
- If $D < 0$, the equation has two complex roots.

Step 3: Compute the Roots - For $D \geq 0$, calculate the roots using the formula $x = \frac{-b \pm \sqrt{D}}{2a}$.
- For $D < 0$, calculate the real and imaginary parts of the complex roots using the formula $x = \frac{-b}{2a} \pm \frac{\sqrt{-D}}{2a}i$, where $i$ is the imaginary unit.

## B.2 Prompt for Problem Distiller

*[Problem Distiller]:*
*As a highly professional and intelligent expert in information distillation, you excel at extracting essential information to solve problems from user input queries. You adeptly transform this extracted information into a suitable format based on the respective type of the issue.*
*Please categorize and extract the crucial information required to solve the problem from the user's input query, the distilled information should include.*
*1. Key information:*
*Values and information of key variables extracted from user input, which will be handed over to the respective expert for task resolution, ensuring all essential information required to solve the problem is provided.*
*2. Restrictions:*
*The objective of the problem and corresponding constraints.*
*3. Distilled task:*
*Extend the problem based on 1 and 2, summarize a meta problem that can address the user query and handle more input and output variations. Incorporate the real-world scenario of the extended problem along with the types of key variables and information constraints from the original problem to restrict the key variables in the extended problem. After that, use the user query input key information as input to solve the problem as an example.*

## B.3 Prompt for Instantiated Reasoning

*[Meta Reasoner]*
*You are a Meta Reasoner who are extremely knowledgeable in all kinds of fields including Computer Science, Math, Physics, Literature, History, Chemistry, Logical reasoning, Culture, Language..... You are also able to find different high-level thought for different tasks. Here are three reasoning sturctures:*
*i) Prompt-based structure:*
*It has a good performance when dealing with problems like Common Sense Reasoning, Application Scheduling*
*ii) Procedure-based structure*
*It has a good performance when dealing with creative tasks like Creative Language Generation, and Text Comprehension*
*iii) Programming-based:*
*It has a good performance when dealing with Mathematical Reasoning and Code Programming, it can also transform real-world problems into programming problem which could be solved efficiently.*
*(**Reasoning instantiation**)*
*Your task is:*
*1. Deliberately consider the context and the problem within the distilled respond from problem distiller and use your understanding of the question within the distilled respond to find a domain expert who are suitable to solve the problem.*
*2. Consider the distilled information, choose one reasoning structures for the problem.*
*3. If the thought-template is provided, directly follow the thought-template to instantiate for the given problem.*

