# OpenReview forum: "Buffer of Thoughts: Thought-Augmented Reasoning with Large Language Models"
_NeurIPS.cc/2024/Conference — NeurIPS 2024 spotlight_

### Official Review · Reviewer_cwbi · 2024-06-27

**Soundness:** 3
**Presentation:** 3
**Contribution:** 3
**Rating:** 7
**Confidence:** 4

**Summary:**

The authors propose a new framework to optimize prompting for various tasks. The frameworks consists of a buffer, where high-level problem-solving templates are stored. A query is first processed by extracting its key aspects.Templates are selected using embeddings of the templates and the extracted key aspects query. The high-level problem-solving templates is then instantiated with the extracted key aspects of the query and its inference is executed. Additionally a buffer manager is introduced to allow updating of the thought template buffer with new, "specialized", thought templates.

**Strengths:**

The paper combines the idea of RAG with a library of templates for solving tasks in order to make automatic prompt engineering more flexible and by using a cache (the library) more efficient, since it avoids more complex exploration. The framework and its components are sufficiently outlined and their description is clear. The writing is easy to follow. The evaluation shows clear advantages over existing approaches, while being executed with modern LLMs.

**Weaknesses:**

Claims from the checklist:
* point 6: It is not clear, what embedding model is used.
* point 8: No sufficient information are provided for the compute resources, especially the local (?) Mixtral execution.

Not clear, how well their approach scales with the number of different thought templates (and not just specialized one). Additionally it is not clear, how well their approach works, if the query (something completely different) does not match any of thought templates in the buffer, as at least some of thought templates seem to specifically designed for the respective evaluation tasks. This is also one of the weaker points, since the authors claim that their approach requires less manually labor from the user than other approaches.

I would have appreciate some examples for the tasks in the appendix, so that it is not necessary to look them up in the respective code/papers.

The language can be polished a little bit more. For example:
* lines 29/30: "which makes it impractical to manually design them task by task" - it is just cumbersome to manually design them, but not impossible
* lines 69/70: "the retrieval-augmented LLM first queries an external database with billion-level tokens [23] for retrieving a subset of the text" - I think that is just how that particular paper implemented RAG, but it is not a general description of the idea behind RAG.
* typos:
  * "enahnced" (line 78)
  * Llama-70B (line 277)

references could also use a little but of work, for example:
* reference [27]: no place of publication
* references are slightly inconsistent:
  * for example one time the abbreviation ICLR is used, otherwise not
  * sometimes the number prefix (for example Eleventh) for conferences is used, sometimes not

**Questions:**

* What embedding model is used for the thought buffer?
* Why it is too computationally expensive to provide error bars for the existing results? It should be possible to compute them locally at least for some of the tasks.
* Figure 3:
  * What is logarithmic time?
  * Why is the inference time of BoT not at least twice that of the Expert baseline, since it needs at least two LLM interactions, where as the Expert baseline only requires one (if I understood that correctly)?

**Limitations:**

The limitations are reasonably explained.

---

> ### Author Rebuttal · Authors · 2024-08-04
>
> *We thank Reviewer cwbi for the positive review and valuable feedback. We are glad that the reviewer found that our proposed BoT is more flexible and more efficient, and our method shows clear advantages over existing approaches. Please see below for our responses to your comments.*
>
>
> **Q1: What embedding model is used?**
>
> **A1:**  In our implementation, we used text-embedding-3-large as the embedding model. We also conduct in-depth experiments about the model size impact for our method. **Please refer to global response for detail.**
>
> **Q2: Information for compute resources?**
>
> **A2:**  As mentioned in Line 214, we use NVIDIA A100-PCIE-40GB GPUs for local model like Llama3-8B and Llama3-70B.
>
> **Q3: How well the approach scale with different thought templates. If query does not match any of thought templates in the buffer, how well this approach works.**
>
> **A3:**
>
> 1. As depicted in Fig.9 and Fig.10, with the number of different thought template increase, the accuracy and reasoning efficiency of BoT steadily improves, it confirms that our method possesses excellent scalability. This is because the buffer manager continually expands the meta-buffer while also utilizing the thought templates obtained from previously solved problems to address subsequent similar problems. The possibility of retrieving suitable thought templates also increases, thus avoiding the need to construct reasoning structures from scratch and thereby enhancing inference efficiency.
>
> 2. As discussed in L160-163, for entirely new or unique problems, our method provides three general coarse-grained thought-templates, which could be assigned considering the distilled task. These coarse-grained thought templates ensure that new tasks follow appropriate reasoning processes. This broad guidance offers LLMs greater flexibility during the generative inference process, enabling them to efficiently address new and unique problems.
>
> 3. Due to character limits, **please refer to Q1 in response to reviewer yiEf** for more examples in demonstrating how well our thought templates work.
>
> **Q4: Language, typos and references**
>
> **A4:**  We sincerely appreciate your attention to the details of our paper. We will carefully address the issues you pointed out regarding language, formatting, and citations, to enhance the professionalism and standardization of our paper.
>
> **Q5: Concern about error bars**
>
> **A5:**
>
> 1.It is indeed possible to compute error bars for our results. However, we chose not to include them in our initial version to maintain consistency with many prior works, such as ToT, Meta Prompting, and PAL, which also do not provide error bars. This approach ensures a unified basis for comparison across different methods.
>
> 2.In response to the reviewer's concern, we have now computed the error bars for our results to enhance the strictness and clarity of our results. The updated results, including error bars, are presented in the table below. We will incorporate these error bars into the final version of our paper to provide a more comprehensive and statistically robust presentation of our results.
>
> | Task|GPT4|GPT4+CoT|Expert|PAL|ToT|GoT|Meta|**BoT (Ours)**|
> |---|----|-------|------|---|---|---|----|--------------|
> | Game of 24|3.0|11.0|3.0|64.0|74.0|73.2|67.0|82.4 ± 1.5|
> | MGSM (avg)|84.4|85.5|85.0|72.0|86.4|87.0|84.8|89.2 ± 1.8|
> | Multi-Step Arithmetic|84.0|83.2|83.2|87.4|88.2|89.2|90.0|99.8 ± 0.2||
> | WordSorting|80.4|83.6|85.2|93.2|96.4|98.4|99.6|100.0 ± 0.0|
> | Python Puzzles|31.1|36.3|33.8|47.3|43.5|41.9|45.8|52.4 ± 1.6|
> | Geometric Shapes|52.6|69.2|55.2|51.2|56.8|54.2|78.2|93.6 ± 2.4|
> | Checkmate-in-One|36.4|32.8|39.6|10.8|49.2|51.4|57.2|86.4 ± 1.7|
> | Date Understanding|68.4|69.6|68.4|76.2|78.6|77.4|79.2|88.2 ± 1.5|
> | Penguins|71.1|73.6|75.8|93.3|84.2|85.4|88.6|94.7 ± 1.2|
> | Sonnet Writing|62.0|71.2|74.0|36.2|68.4|62.8|79.6|80.0 ± 0.4|
>
> As shown in the table, our method achieves stable performance, which further proofs that our method is effective, stable and robust.
>
> **Q6: What is logarithmic time?**
>
> **A6:** We perform a logarithmic operation on the obtained time, i.e., Logarithm time = $\ln(t)$. The reason why we do this is that the data range of the inference time between different methods is board, in such cases, the readability and effectiveness of the histogram in conveying information are compromised. By applying logarithmic transformation, the differences in bar heights within the histogram are reduced, rendering the chart more aesthetically pleasing and the information more clearly presented.
>
> **Q7: Why is the inference time of BoT not at least twice that of the Expert baseline?**
>
> **A7:** As mentioned in Q6, the formula for calculating the multiple of inference time should be $e^{\ln(t_1) - \ln(t_2)}$, rather than direct comparison. Furthermore, despite our method involving collaboration across multiple components, the inference time required by BoT for certain problems does not significantly exceed that of the baseline. Because our method transforms some reasoning tasks (such as Game of 24 and checkmate in one) by simplifying the multi-step reasoning or heuristic search process into generating a segment of code capable of solving the problem. Specifically, in the expert baseline, solving a problem from Game of 24 need to experiment with multiple combinations and conduct calculations, which is time consuming. On the contraty, our method transform this process into generating few lines of code and execute the code to get the answer,  which requires significantly less time compared to the expert baseline, demonstrating the superiority of our approach in addressing certain complex problems.

---

> > ### Comment · Reviewer_cwbi · 2024-08-13
> >
> > I have read the rebuttal to my review as well as the rebuttals and comments for the other reviewers.
> >
> > Q6: Regarding Fig. 3, I find the figure rather misleading or at least hard to read, especially with current y axis label, which suggests that the time is seconds. I suggest using a logarithmic scale of the y axis instead of logarithmic time.
> >
> > Q7: But then shouldn't you use a better baseline? I mean clearly Game of 24 can be solved by a simple Python script.

---

> ### Author Response · Authors · 2024-08-13
> **Gentle Reminder**
>
> Dear Reviewer cwbi,
>
> We greatly appreciate the time and effort you have invested in reviewing our paper. Your thoughtful questions and insightful feedback have been invaluable. In response to your queries, we provide more explanations about the effectiveness of our proposed approach, and additional analysis experiments.
>
> If you have any further questions, please feel free to ask. Thank you once again for your invaluable contribution to our research.
>
> Warm regards, The Authors

---

> ### Author Response · Authors · 2024-08-13
> **Response to Reviewer cwbi**
>
> Thank you for your reply, and we sincerely appreciate your suggestions! Following your suggestion, we will using a logarithmic scale of the y axis instead of logarithmic time in Fig.3 for the final version. Actually, in Fig.3, we have compared our method with PAL (Program-aided Language Models) which utilizes a python script to solve problem. And from the results in Fig.3, we can find that our BoT is faster than PAL because BoT retrieves proper thought templates for accelerating our reasoning process without reasoning from scratch.
>
> If you have any further questions, please feel free to ask. Thank you once again for your invaluable contribution to our research.
>
> Warm Regards,
>
> The Authors

---

### Official Review · Reviewer_upCD · 2024-07-09

**Soundness:** 3
**Presentation:** 3
**Contribution:** 2
**Rating:** 6
**Confidence:** 4

**Summary:**

This paper proposes a novel reasoning procedure for LLM, named "buffer of thoughts". The core idea is that for each task, first extract a template describing how the task should be solved, then store all these templates in a "buffer". As the LLM receives a new query, a retrieval procedure is applied to extract the most relevant task template out of the buffer. The template is then instantiated for this specific query, which becomes a concrete guideline that instructs the model how to solve the new query. The paper demonstrates that this procedure outperforms baselines like tree-of-thought and graph-of-thought significantly on some reasoning tasks.

**Strengths:**

The novelty of the proposed method "buffer of thought" is a good contribution of the paper which may significantly improve LM's reasoning ability for some tasks.

**Weaknesses:**

Generalizability and robustness of the proposed approach needs to be further verified, as the introduction of a hierarchical reasoning procedure (abstract template + instantiation) may create more noise and make it less reliable for novel tasks.

**Questions:**

1. The success of the proposed approach critically depends on the quality of the automatically induced template. While the paper showed empirical performance on downstream tasks, it remains unclear how good the templates themselves are. It would be great to make a comparison between the automatically generated task templates with manually prepared templates for some tasks.

2. It remains unclear to me how the template for a specific task category (say maths) can potentially be revised and improved, as the model sees more and more examples from that task. Although the paper described briefly how the buffer can be dynamically updated (line 182), I don't quite understand how the template is updated automatically for higher accuracy as the model receives more examples (this seems to be a verbalized gradient descent procedure, but no quality improvement is guaranteed).

3. What are the embedding models you used for retrieval purposes? Also it would be great to give some analysis on the impact of retrieval quality on downstream task performance, as finding the right template is critical.

4. In Eq 5, it seems whether the template should be replaced should depend upon the template quality, instead of its similarity with the embedding vector.

5. How quickly can this approach adapt to an unseen task? Would one or two examples from new task be enough to achieve reasonable accuracy? It would be great to give some analysis to the adaptability and continual learning ability of the proposed approach.

**Limitations:**

The paper properly addressed limitations and impact of the work.

---

> ### Author Rebuttal · Authors · 2024-08-04
>
> *We thank Reviewer upCD for the positive review and valuable feedback. We are glad that the reviewer found that our BoT makes good contributions to the improvements of LLMs, and our method significantly improves LM's reasoning ability in various tasks. Please see below for our responses to your comments.*
>
> **Q1: Generalizability and robustness of the proposed approach.**
>
> **A1:**
>
> 1. Breaking down a complex problem into step-by-step solutions has proven to be an effective way to enhance accuracy and robustness, as demonstrated by ToT, GoT, and Meta-prompting. Moreover, Our BoT additionally introduces a problem distiller to extract the core problem for input task and simplifies the problem-solving procedure by retrieving proper thought templates. Such high-level thought templates discard unnecessary specific details of problem-solving processes and can be instantiated with different problems adaptively, which effectively reduce possible noises caused by previous hierarchical reasoning methods (ToT, GoT).
>
> 2. The empirical results have also proved the generalizability and robustness of our BoT. The experiments of Fig.4 demonstrate that our BoT can maintain a robust reasoning process and reduce more noises compared to previous methods when faced with different tasks. And the experiments of Fig.9 reveal that our BoT can generalize to more novel tasks and continually improve the performance.
>
> **Q2: Comparison between automatically generated task templates with manually prepared templates.**
>
> **A2:**
>
> 1. We appreciate the reviewer's insightful comment regarding the quality of the automatically generated templates. To address this, we conducted comprehensive experiments to compare the automatically-generated templates with manually-designed templates in MATH dataset. Please refer to **the PDF in global response** for more detailed results.
>
> 2. The additional experiments showcase that the automatically-generated templates ensure consistency, efficiency and scalability for more accurate and effective problem-solving, also demonstrated by the results in Fig.9 and 10.
>
> **Q3: How template can be revised and improved?**
>
> **A3:**
>
> 1. Since there is no suitable metric to evaluate the quality of thought templates and BoT is a training-free reasoning fraemwork, we **do not optimize the previously accumulated thought templates but instead add new thought templates**. To be specific, we calculate the similarity between new thought templates and those in the meta-buffer to avoid redundancy and repetition. Using powerful LLMs like GPT-4, the quality of the extracted thought templates is relatively high, as demonstrated by our qualitative analysis in Fig.9 and 10.
>
> 2. As for the reason why the accuracy increase with more examples received in Fig.9, we have discussed this in Line 465-477. That is because when in the first round, the meta-buffer is empty and the buffer manager is still accumulating thought templates. With the accumulation of thought templates, BoT gradually enhances its ability **by utilizing the thought-templates obtained from previously solved problems to help** addressing subsequent similar problems, thus contributing to the increase in accuracy.
>
> 3. We sincerely thank you for your insightful suggestion. In future work, we plan to design a metric to evaluate the quality of thought templates, allowing for dynamic optimization of the accumulated thought templates. This enhancement may further improve the accuracy and robustness of our method.
>
> **Q4: What are the embedding models used for retrieval ? And analysis on the impact of retrieval quality.**
>
> **A4:** We used text-embedding-3-large as the embedding model. For better understanding, we here provide experimental analysis on the impact of retrieval quality on downstream task performance using three different-sized embedding models. Due to the page limits, **please refer to the table and analysis in global response**.
>
> **Q5: In Eq.5, whether we should replace the template based on similarity.**
>
> **A5:** It is noted that our update process **only includes adding new thought templates, instead of replacing or optimizing** existing ones . As mentioned in above A3, there is no suitable metric to evaluate the quality of thought template and BoT is a training-free reasoning framework. For future work, we will try to effectively evaluate the quality of thought templates, thereby enabling the dynamic optimization of meta buffer.
>
>
> **Q6: How quickly can this approach adapt to an unseen task? Number of examples to achieve reasonable accuracy? Analysis of adaptability and continual learning ability of BoT.**
>
> **A6:**
>
> 1. The adaptation is related with task diversity. For tasks with low diversity, such as "Game of 24", our approach can adapt very quickly.  For tasks with higher diversity, such as those in MATH dataset, which includes a wide range of problem types (e.g., algebra, geometry...), more examples are required to accumulate a comprehensive set of thought templates. This is necessary to cover the various subdomains of the problems.
>
> 2. For tasks with low diversity, one or two examples are often sufficient to achieve reasonable accuracy. For tasks with higher diversity, a larger set of examples is necessary as mentioned above. This ensures that our method remains robust and generalizable across a wide range of problem types.
>
> 3. As shown in in Fig.9 and Fig.10 of Appendix, we conduct ablation study on buffer-manager, which demonstrate the extradionary adaptability and continual learning ability of BoT. With more examples received and high-level thought templates accumulated in each round, there is significant improvement in both the overall performance and reasoning efficiency of the model. We have discussed the underlying reasons for these improvements in A3. This continual learning capability is crucial for ensuring that BoT can effectively adapt to new tasks over time, maintaining the efficiency and robustness of our method.

---

> > ### Comment · Reviewer_upCD · 2024-08-14
> >
> > Thank you for your detailed response, it addressed some of my concerns. I have updated my rating to 6.

---

> ### Author Response · Authors · 2024-08-13
> **Gentle Reminder**
>
> Dear Reviewer upCD,
>
> We greatly appreciate the time and effort you have invested in reviewing our paper. Your thoughtful questions and insightful feedback have been invaluable. In response to your queries, we provide further explanations about the generalizability and robustness of our proposed approach, and more details about our thought-retrieval procedure.
>
> If you have any further questions, please feel free to ask. Thank you once again for your invaluable contribution to our research.
>
> Warm regards, The Authors

---

### Official Review · Reviewer_yiEf · 2024-07-11

**Soundness:** 3
**Presentation:** 3
**Contribution:** 3
**Rating:** 7
**Confidence:** 4

**Summary:**

This paper proposes a new approach called Buffer of Thoughts (BoT) to improve LLMs reasoning abilities. BoT addresses this by creating a "meta-buffer" that stores general problem-solving “thought” templates across different tasks. When a new query is given as input, BoT retrieves a relevant thought from the meta-buffer and tailors it to the specific situation. The method also includes a "buffer-manager" to keep the meta-buffer up-to-date and effective as the LLM encounters new challenges for scalability and stability purposes. The authors perform experiments on different reasoning tasks that showed significant improvements with BoT compared to previous methods.

**Strengths:**

- The framework is clearly explained and the paper is easy to follow
- The empirical evaluation is thorough
- The improvement for some tasks is notable.
- The idea of a library of thoughts templates is sound and novel

**Weaknesses:**

- BoT seems to work well for problems with existing templates in the meta-buffer. However, entirely new or unique problems might not have a relevant template for BoT to adapt, potentially hindering its ability to solve them effectively.
- What happens in the case of unsolvable queries? Does the method allows for uncertainty based abstention?

**Questions:**

see weaknesses.

**Limitations:**

limitations have been discussed in the paper.

---

> ### Author Rebuttal · Authors · 2024-08-04
>
> *We thank Reviewer yiEf for the positive review and valuable feedback. We are glad that the reviewer found that the proposed thought template and meta-buffer is novel, and our method achieve notable improvement across various tasks. Please see below for our responses to your comments.*
>
> **Q1: How to solve entirely new or unique problems effectively?**
>
> **A1:**
>
> 1.  As discussed in Line 160-163, for entirely new or unique problems, our method provides **three pre-designed coarse-grained thought templates** (in  Appendix A.3) that can be instantiated based on the distilled task information. These templates offer a good starting point for reasoning when the relavant thought template does not exist.
>
> 2.  Additionally, these coarse-grained thought templates ensure that new tasks follow appropriate inference processes. This general guidance offers LLMs greater flexibility during the reasoning process, enabling them to efficiently address new and unique problems.
>
> 3.  During the cold-start reasoning process mentioned above, the buffer manager distills and summarizes new thought templates that can be used to solve similar future problems effectively. This adaptive process ensures that our method remains effective and robust, and could continually adapt to new problems.
>
> 4. To give a comprehensive illustration of this reasoning process for new or unique problems, we use the Game of 24 and solving a quadratic equation as examples below:
>
> **Game of 24**
>
> Input problem:
>
> Let's play a game called 24. You'll be given four integers, and your objective is to use each number only once, combined with any of the four arithmetic operations (addition, subtraction, multiplication, and division) and parentheses, to achieve a total of 24. For example, if the input is 4, 7, 8, and 8, the output could be 7 * 8 - 4 * 8 = 24. You only need to find one feasible solution!
>
> **Problem Statement:**
>
> Given four integers 4, 7, 8, and 8, use each number exactly once and the four arithmetic operations (addition, subtraction, multiplication, division) to form an expression that equals 24. Parentheses can be used to define the order of operations.
>
> Analysis:
>
> **Step 1: Problem Understanding**: We need to find a combination of the four given integers and arithmetic operations that results in 24.
>
> **Step 2: Input Extraction**: The four integers are 4, 7, 8, and 8.
>
> **Step 3: Output Requirements**: The output should be a valid mathematical expression that equals 24.
>
> **Step 4: Constraints**:
>
> - Each number must be used exactly once.
> - Only the four arithmetic operations can be used.
> - Parentheses can be used to define the order of operations.
>
> **Step 5: Approach**:
>
> - Generate all permutations of the numbers.
> - Generate all possible combinations of arithmetic operations.
> - Evaluate all possible expressions formed by combining the numbers and operations with different parenthetical groupings.
> - Check if any of these expressions evaluate to 24.
>
> **Transformed Python Code:**
>
> ```python
> ... Due to character limit, we are unable not show the code here, you can refer to A.1.5 in our paper ...
> ```
>
> **Solving a quadratic equation**
>
> **Input Problem:**
>
> Solving a quadratic equation：
>
> $2x^2-4x-6 = 0$
>
>
> **Solution:**
>
> Step 1: Identify the coefficients \(a\), \(b\), and \(c\) from the quadratic equation $(2x^2 - 4x - 6 = 0)$:
>
> - \(a = 2\)
> - \(b = -4\)
> - \(c = -6\)
>
> Step 2: Calculate the discriminant \(D\):
>
> $D = b^2 - 4ac = (-4)^2 - 4 \cdot 2 \cdot (-6) = 16 + 48 = 64$
>
> Step 3: Determine the nature of the roots:
>
> - Since \(D = 64 > 0\), the equation has two distinct real roots.
>
> Step 4: Compute the roots using the quadratic formula:
>
> $x = \frac{-b \pm \sqrt{D}}{2a} = \frac{-(-4) \pm \sqrt{64}}{2 \cdot 2} = \frac{4 \pm 8}{4}$
>
> - Calculate the two roots:
>
>   $x_1 = \frac{4 + 8}{4} = \frac{12}{4} = 3$
>
>
>   $x_2 = \frac{4 - 8}{4} = \frac{-4}{4} = -1$
>
> **Answer:**
> The solutions for the quadratic equation $(2x^2 - 4x - 6 = 0)$ are:
>
> $x_1 = 3 \quad \text{and} \quad x_2 = -1$
>
> In summary, as we mentioned in Line 160-163 and the example in Appendix A.3, we can effectively solve entirely new or unique problems. The example above further showcase the robustness of our method when encountering new tasks.
>
> **Q2: What happens in the case of unsolvable queries? Does the method allows for uncertainty based abstention?**
>
> **A2:**
>
> 1. For possible unsolvable queries, our method attempts to resolve the issue during the instantiation process. Subsequently, an additional inspector reviews both the problem and the reasoning process. If any issues are detected, the unreasonable parts are identified and handed over to the reasoner for revision. If it is concluded that the problem is unsolvable, the reasoning process is terminated immediately to prevent any pollution of the meta-buffer.
>
> 2. Due to the page limits, we did not discuss this scenario in our paper, and there were very few unsolvable queries in our experiments. However, we have implemented the inspector mechanism. We will add this part in the final version to provide a comprehensive overview of our approach.

---

> > ### Comment · Reviewer_yiEf · 2024-08-11
> > **Response to rebuttal**
> >
> > I appreciate the authors' responses to my questions. I maintain my current score and evaluation.

---

### Official Review · Reviewer_hZUQ · 2024-07-13

**Soundness:** 3
**Presentation:** 3
**Contribution:** 3
**Rating:** 8
**Confidence:** 4

**Summary:**

The paper introduces "Buffer of Thoughts" (BoT), a novel framework designed to improve the reasoning abilities of large language models (LLMs) by incorporating a 'thought-augmented' approach. This framework uses a component called 'meta-buffer' to store high-level thoughts—concise, distilled reasoning strategies from various problem-solving instances—which can be dynamically retrieved and adapted to new tasks to facilitate efficient reasoning. This approach significantly enhances the accuracy, efficiency, and robustness of LLMs across multiple reasoning-intensive tasks.
1. Novel Framework: BoT innovates a thought-augmented reasoning mechanism that leverages previous problem-solving insights to aid new reasoning tasks, reducing the need for generating reasoning paths from scratch.
2. Meta-Buffer: Introduces a storage system that holds distilled high-level thoughts, allowing for rapid adaptation to different tasks by retrieving and instantiating these thoughts as needed.
3. Buffer Manager: A dynamic management system that updates the meta-buffer based on newly encountered tasks, continuously improving the system’s reasoning capacity.
4. Empirical Validation: The paper reports extensive testing across 10 complex tasks, demonstrating significant performance improvements over state-of-the-art methods, including improvements of 11% on Game of 24, 20% on Geometric Shapes, and 51% on Checkmate-in-One, while also reducing the computational cost compared to traditional multi-query prompting methods.

**Strengths:**

1. **Improved Reasoning Accuracy:** BoT significantly enhances the reasoning accuracy of large language models by leveraging distilled high-level thoughts, allowing the models to approach problems with a pre-formed strategy that has been proven effective across tasks.
2. **Computational Efficiency:** The thought-augmented reasoning approach minimizes the need for complex and iterative query processes typical in multi-query prompting systems. By reusing structured thought templates, BoT reduces the computational overhead, leading to faster reasoning times.
3. **Robustness Across Tasks:** BoT exhibits a robust performance over a range of different and challenging tasks. This is attributed to the system's ability to adapt high-level reasoning thoughts to new problems, ensuring consistent performance without the need to tailor the system for specific tasks.

**Weaknesses:**

1. **Dependence on Quality of Distilled Thoughts:** The effectiveness of BoT hinges significantly on the quality of the distilled thoughts stored in the meta-buffer. If the initial thoughts distilled from problem-solving processes are not sufficiently generalizable or are too simplistic, they may not provide the necessary depth for complex reasoning tasks.
2. **Scalability and Maintenance of the Meta-Buffer:** The paper introduces a dynamic system for updating the meta-buffer but does not deeply explore the long-term scalability and maintenance challenges associated with continuously growing and updating this repository. Managing an ever-expanding set of thought templates could lead to efficiency issues or dilution of useful thoughts.
3. **Risk of Overfitting to Distilled Thoughts:** There is a potential risk that the LLM might overfit to the specific styles or patterns of reasoning encapsulated in the thought templates, especially if these templates are derived from a limited set of problem-solving instances.

**Questions:**

1. As the meta-buffer grows with more distilled thoughts, how does its size impact the retrieval time and overall performance of the model?
2. In cases where multiple applicable thoughts could be retrieved from the meta-buffer, how does BoT prioritize or choose among conflicting reasoning strategies?
3. Given the reliance on previously distilled thoughts, how does BoT handle situations where the foundational thoughts might be based on incorrect or outdated information?

**Limitations:**

One of the primary limitations highlighted in the discussion of the Buffer of Thoughts (BoT) framework is its dependency on the initial quality of the meta-buffer. The performance of BoT is contingent upon the initialization of this buffer. If initialized with a weak model, the distilled thoughts stored in the meta-buffer may not be of sufficient quality or depth to facilitate effective reasoning across diverse tasks.

---

> ### Author Rebuttal · Authors · 2024-08-04
>
> *We thank Reviewer hZUQ for the positive review and valuable feedback. We are glad that the reviewer found that the proposed framework is novel, and greatly enhances the accuracy, efficiency, and robustness of LLMs across multiple reasoning tasks. Please see below for our responses to your comments.*
>
> **Q1: Dependence on Quality of Distilled Thoughts.**
>
> **A1:**
>
> 1. We address this by utilizing pre-designed prompt and leveraging state-of-the-art LLMs such as GPT-4, which are capable of generating high-quality, high-level thought templates that strike a balance between abstraction and specificity.
>
> 2. To demonstrate the adaptability and robustness of our method, we provide examples from two distinct datasets: GSM8K (grade school math word problems) and MATH (challenging high school math competition problems).
>
> **Example 1: GSM8K**
>
> **Question**: A robe takes 2 bolts of blue fiber and half that much white fiber. How many bolts in total does it take?
>
> **High-level Thought Template:**
>
> Step 1: Identify the amount of each type of material required.
>
> Step 2: Calculate any dependent quantities based on the given relationships.
>
> Step 3: Sum all the quantities to find the total amount of material needed.
>
> **Example 2: MATH**
>
> **Question**: At a school, all 60 students play on at least one of three teams: Basketball, Soccer, and Mathletics. 8 students play all three sports, half the students play basketball, and the ratio of the size of the math team to the size of the basketball team to the size of the soccer team is $4:3:2$. How many students at the school play on exactly two teams?
>
> **High-level Thought Template:**
>
> Given:
>
> Total number of students: T
>
> Number of students in each team: A, B, C
>
> Number of students playing all three sports: y
>
> Step 1: Identify the total number of students T and the number of students in each team A, B, and C.
>
> Step 2: Recognize the overlapping memberships and set up the equation:
> $$
> A + B + C = T + x + 2y
> $$
>
> Step 3: Substitute the known values for A, B, C, T, and y into the equation.
>
> Step 4: Solve for x, the number of students playing exactly two sports:
> $$
> x = (A + B + C) - T - 2y
> $$
>
> These examples illustrate that our method can adaptively extract high-quality thought templates for problems of different complexity. For simpler problems from GSM8K, the high-level thought template is straightforward, while for more complex problems from MATH, the high-level thought template is correspondingly complicated. This adaptability and generalizability showcase the robustness of our approach.
>
>
> **Q2: Risk of Overfitting to Distilled Thoughts.**
>
> **A2:**
>
> 1. Our thought templates are designed to be high-level and abstract, which do not contain specific reasoning details from previous problems. This abstraction ensures the thought templates provide general guidance that would not cause overfitting problems of LLMs.
>
> 2. In scenarios where the meta-buffer contains thought templates derived from a limited set of problem-solving instances that differ significantly from the current problems, we reinstantiate the reasoning process using our manually pre-designed coarse-grained thought templates, as mentioned in Line 450. This approach avoids applying an incorrect type of thought template from the meta-buffer, thereby further mitigating the risk of overfitting.
>
> 3. As we mentioned above, our approach ensures that the model retains the flexibility needed to handle a wide range of tasks effectively while maintain adaptability and robustness across various problems.
>
>
> **Q3: Size impact on retrieval time and overall performance. Concern about the Scalability and Maintenance.**
>
> **A3:**
>
> 1. Many different problems share similar solutions that could be instantiated from the same high-level thought template. Consequently, even as the meta-buffer grows with more distilled thoughts, the number of unique thought templates remains relatively small as demonstrated by the template distribution of Fig.5. This ensures that our meta-buffer is a lightweight library, and its size has little impact on retrieval time. Moreover, a bigger size of meta buffer would improve the overall performance and the reasoning efficiency of the model as in Fig.9 and Fig.10, with empirical analysis in Line 465-477.
>
> 2. Many problems share the same reasoning patterns, we only need to save the high-level thought templates distilled from the solutions of various problems. This approach keeps the size of the meta-buffer relatively small, ensuring its scalability and making it easy to maintain.
>
>
> **Q4: How does BoT prioritize among conflicting reasoning strategies?**
>
> **A4:**
>
> As illustrated in Line 134-144, for each thought template, we have a description $D_{T_i}$ that specifies the types of problems could be applied. BoT uses the embedding similarity between the distilled problem $x_d$ and $D_{T_i}$ to find the most suitable thought template, as mentioned in Eq.2. We only **choose the thought template with the highest similarity** to the current problem. This ensures that the chosen template aligns closely with the current problem, thereby prioritizing the most relevant reasoning strategy.
>
> **Q5: How to handle situations when thoughts are based on incorrect or outdated information?**
>
> **A5:**
>
> 1. The thought templates derived from the problem-solving process are high-level and abstract, which means they **do not contain specific reasoning details or factual information**. This abstraction helps to ensure that the templates provide robust and generalizable guidance without being affected by the incorrect examples or outdated data.
>
> 2. Compared to conventional methods that rely on in-context examples or specific databases, our approach leverages high-level thought templates to eliminate detailed reasoning steps. This abstraction helps to avoid potential errors that might be presented in more detailed templates, thereby enhancing the reliability and robustness of the reasoning process.

---

> > ### Comment · Reviewer_hZUQ · 2024-08-10
> > **Official Comment by Reviewer hZUQ**
> >
> > Thank you for the comprehensive response and additional experiments provided. I find this to be a very interesting and excellent piece of work and look forward to meeting the authors at the conference. I have increased my score from 7 to 8, and I hope the authors will incorporate these suggestions into the final manuscript.

---

> > > ### Author Response · Authors · 2024-08-11
> > > **Thank you for your support**
> > >
> > > Thank you very much for raising the score! We sincerely appreciate your valuable comments and the time and effort you put into reviewing our paper. We will make sure to incorporate these suggestions into the final manuscript.
> > >
> > > Warm Regards,
> > > The Authors

---

### Author Rebuttal · Authors · 2024-08-04

Global response

We sincerely thank all the reviewers for their thorough reviews and valuable feedback. We are glad to hear that the proposed framework is novel (all reviewers) and effective (all reviewers) in enhancing the reasoning abilities of LLMs, the paper is well-written and easy to follow (reviewer yiEf and cwbi), and the performance improvements demonstrated in experiments are promising (all reviewers).

Here, we want to highlight the main contributions and novelties of our proposed framework, **"Buffer of Thoughts" (BoT)**:

**Thought-Augmented Reasoning Mechanism:** BoT introduces a novel framework that leverages previously distilled high-level thoughts to assist LLM reasoning tasks. This reduces the need for generating reasoning paths from scratch, significantly enhancing accuracy, efficiency, and robustness.

**Meta-Buffer and Buffer Manager:** BoT incorporates a meta-buffer to store distilled high-level thoughts and a dynamic buffer manager to continuously update the meta-buffer based on new tasks. This ensures scalability and continuous improvement of the LLM reasoning system.

**Empirical Validation:** Extensive testing across 10 complex tasks demonstrates significant performance improvements over state-of-the-art methods, including substantial accuracy improves and reduced computational costs compared to traditional multi-query prompting methods.

We summarize our responses to the reviewers' comments as follows:

* We additionally provide more examples to show the quality of our high-level thought template. (reviewer hZUQ, yiEf and upCD).

* We provide more examples to demonstrate the instantiation and reasoning process for new problems, and give detailed analysis about the adaptability and continual learning ability of our method. (reviewer yiEf, upCD and cwbi).

* We further conduct the experiment to compare the quality of automatically-generated thought template and manually-designed template in response pdf. (reviewer hZUQ and upCD)

* We provide more quantitative comparisons and analysis for the impact of different-sized embedding model in the table below (reviewer upCD and cwbi).

  * The results indicate that stronger encoding capabilities lead to higher accuracy. The larger model (text-embedding-3-large) extracts more informative embeddings, improving retrieval accuracy and overall BoT performance.

  * The impact of embedding model is not significant. Even with the small text-embedding-ada-002 model, we still achieve higher accuracy than other methods. This is because we use the distilled problem $x_d$ and the thought template description ($D_T$) for similarity computation. Both $x_d$ and $D_T$ are concise sentences, with simple but critical semantic structures, allowing even weaker encoders to handle them effectively. Thus, our method remains robust and generalizable across different-sized encoder, as demonstrated by the experiment results.

|Task (accuracy)|text-embedding-3-large+BoT|text-embedding-3-small+BoT|text-embedding-ada-002+BoT|GPT4|ToT|Meta Prompt|
|---|:---:|:---:|:---:|:---:|:---:|:---:|
|Game of 24|82.4|81.8|81.0|3.0|74.0|67.0|
|MGSM|89.2|88.7|87.9|84.4|86.4|84.8|
|Word Sorting|100.0|100.0|99.8|80.4|96.4|99.6|


We reply to each reviewer's questions in detail below their reviews. Please kindly check out them. Thank you and please feel free to ask any further question.

---

### Decision · Program_Chairs · 2024-09-25

**Decision:**

Accept (spotlight)

**Comment:**

The paper introduces "Buffer of Thoughts" (BoT), a novel framework designed to improve the reasoning abilities of large language models (LLMs). This framework uses a component called 'meta-buffer' to store high-level problem-solving strategies, which can be dynamically retrieved and adapted to new tasks to facilitate efficient reasoning. The method also includes a "buffer-manager" to keep the meta-buffer up-to-date and to support effective scaling. The paper also presents experiments on a range of reasoning tasks showing significant improvements over other methods.

Most reviewers regard the paper as well written and all consider the central idea of buffer of thoughts to be sound, novel and effective. The idea has fairly general applicability and makes justified and substantiated claims regarding computational efficiency and robustness with respect to other recent methods. The paper appears to include a thorough empirical evaluation, with the proposed method performing significantly better than existing methods on some tasks.

There is a question regarding how well the system will perform over a long time-scale: whether it has issues such as over-specialisation, and how well this approach scales with the number of different thought templates. There are other questions about how dependent the system is on a good initial library of thoughts, how effective the system is when it encounters a task for which no good template exists in the buffer, and how it behaves when it encounters unsolvable queries. Nonetheless, this work appears to represent a high quality, novel contribution and may be of significant interest as it relates to a highly active domain of inquiry.